# Biocompatibility Profile and In Vitro Cellular Uptake of Self-assembled Alginate Nanoparticles

**DOI:** 10.3390/molecules24030555

**Published:** 2019-02-03

**Authors:** Pei Zhang, Shirui Zhao, Yaoyao Yu, Huan Wang, Yan Yang, Chenguang Liu

**Affiliations:** 1College of Marine Life Sciences, Ocean University of China, Qingdao 266003, China; zhangpei8877@126.com (P.Z.); mengzi58@snnu.edu.cn (S.Z.); yany@ouc.edu.cn (Y.Y.); 2Department of Life Science, Luoyang Normal University, Luoyang 471934, China; paopaoyu7545@126.com (Y.Y.); huanw0209@163.com (H.W.)

**Keywords:** alginate, degree of substitution, nanoparticles, cytotoxicity, cellular uptake

## Abstract

Polymeric nanoparticles could offer promising controlled drug delivery. The biocompatibility is of extreme importance for future applications in humans. Self-assembled polymeric nanoparticles based on phenylalanine ethyl ester (PAE)-modified alginate (Alg) had been successfully prepared and characterized in our lab. However, little is known about their interaction with cells and other biological systems. In this study, nanoparticles (NPs) based on PAE-Alg conjugates (PEA-NPs) with different degree of substitution (DS) were prepared and investigated. Our results showed that PEA-NPs had no effects on the proliferation of the human intestinal epithelial Caco-2 cells at concentrations up to 1000 μg/mL. Furthermore, the in vitro cellular uptake profile of PEA-NPs, concerning several parameters involved in the application of therapeutic or diagnostic NPs, such as NPs concentration, time and temperature, was described. Different NPs have been adopted for cellular uptake studies and the NPs internalized into Caco-2 cells were quantified. Cellular uptake efficiency could reach 60% within 4 h. PEA-NPs also showed greater cell permeability than oleoyl alginate ester nanoparticles (OAE-NPs) previously prepared in our lab. Our studies reveal that NPs based on PEA conjugate are promising nanosystems for cellular delivery.

## 1. Introduction

In the last two decades it has been found that amphiphilic block or graft copolymers can form self-assembled, nanosize aggregates [1]. In aqueous solution these micelle-like particles consist of an inner hydrophobic core and an outer hydrophilic shell [2,3]. Comprehensive research has been conducted on the feasibility for self-assembled nanoparticles (NPs) to be used for targeted drug delivery and other biomedical applications, such as suitable properties for potential use PDT [4], affecting immune responses [5,6], delivery of proteomics or genomics used in ligand-targeted therapeutics [7], delivery of anticancer and antifungal drugs approved for clinical use [8]. Because of their special physicochemical properties, such as size and surface area, self-assembled NPs have shown great advantages in targeted therapy and delivery [9]. For instance, poly(ethylene glycol)-α-lipoamino moiety (PEG_5000_-LAA) conjugate can not only protect encapsulated substances, but also modulate physicochemical characteristics of drug release [10]. However, the application of NPs is still facing major cytotoxicity problem induced by NPs or their degradation products. Researches to improve biocompatibility of NPs still deserve much attention [11]. Francesco synthesized core-shell imprinted NPs which has good biocompatibility, paving the way for their in vivo application [12]. Liu successfully prepared OCMCS nanoparticles via self-assembly which was no cytotoxicity in vitro MTT assay and in vivo toxicity study, suggested that OCMCS nanoparticles might be employed as a potential approach for oral delivery of macro-molecular drugs [13].

Amphiphilic polysaccharide derivatives have been widely used for targeted drug delivery and other biomedical applications because of their good cytocompatibility and biodegradability [14], such as carboxymethyl chitosan [15], hyaluronic acid [16] and alginate [17]. Alginate (Alg) is a natural polymer with good biocompatibility and has the potential to form self-assembled NPs after hydrophobic modification. Wang have investigated OAlg-based nanoparticles in the self-assembled films not only result in porous nanostructures similar to natural ECM, but also preserve the activity and realize the sustained release of Bone morphogenetic protein-2 [18]. Kolya have prepared gold nanoparticles as antibacterial agents by using graft copolymer based on sodium alginate, dimethylacrylamide and acrylic acid [19]. Yu have prepared the photosensitizer- encapsulated amphiphilic sodium alginate derivative nanoparticles to enhance phototoxicity in the photodynamic therapy of pancreatic cancer [20]. Song fabricated magnetic alginate/CHI layer-by-layer nanoparticles to deliver curcumin for treating with breast cancer cells [21]. The biocompatibility and cellular uptake characteristics of alginate-based NPs are of great importance for biomedical applications [22,23]. The effective cellular uptake profile of NPs may influence its performance as a drug or nutrient carrier [24]. On one hand, cellular uptake efficiency in specific tissue has significant impacts on the therapeutic efficacy because efficient delivery of therapeutic cargo into target cells is critical for drugs that act intracellularly [25]. On the other hand, small intestinal cellular uptake efficiency also influences the bioavailability of the drug and finally affects the effectiveness [26]. Caco-2 cell line, which possesses similar morphology, transportation system, and permeability characteristic to small intestinal cells, has been widely used as in vitro model for drug uptake and transportation research.

There are many challenges in drug delivery research such as standard methods and procedures to produce polymer NPs, highest uptake and lowest cell death levels [27]. In a previous study, we reported that Alg hydrophobically modified with oleic acid (oleoyl alginate ester, OAE) can load water insoluble vitamins into the hydrophobic cores of nanosized particles. OAE nanoparticles presented enhanced small intestinal uptake and transportation, exhibiting great potential in nutrient delivery applications [28]. In this research, we constructed a nanosized system with biocompatible natural polysaccharide and an essential amino acid ester. The biocompatibility and cellular uptake profiles of this nanocarrier were evaluated. It suggests that the Alg modified with hydrophobic PAE could facilitate the self-assembly of the constructed NPs and enhance the cellular uptake of the NPs due to the interaction between PAE and cell membrane.

## 2. Results

### 2.1. Characterization of PEA Nanoparticles

The self-aggregated PEA nanoparticles were constructed by sonication in aqueous condition. The mean hydrodynamic diameters (which determined by DLS, Figure 1B) of the particles decreased from 425.3 ± 5.7 nm to 226.7 ± 4.0 nm with increased DS of PEA conjugate (PEA1 = 3.49, PEA2 = 4.18 and PEA3 = 4.67) [27]. This might be explained by the stronger intra- and intermolecular hydrophobic interaction with increasing DS [29]. As shown in Figure 1A, the TEM observation showed that PEA-NPs had spherical structure with good dispersion. Meanwhile, the mean diameter and size distribution of PEA-NPs appeared to be similar with the DLS results [16].

### 2.2. Cytocompatibility of PEA-NPs

The viability of Caco-2 cells after incubation with PEA-NPs for 48 h was shown in Figure 2. The cell viability in each group was above 95% with the concentration of PEA-NPs ranging from 50 to 1000 μg/mL, demonstrating that PEA-NPs were nontoxic even at high concentration. Alginate is a natural polysaccharide with good biocompatibility and PAE is a hydrophobic segment and chosen as a non-toxic amino acid [30,31]. Thus, the PEA products showed good cytocompatibility. There was also no significant difference in cell viability for PEA with different DS which implies that the PAE modification did not affect the biocompatibility of Alg. However, high concentrations of nanoparticles are still toxic to cells over a long period of time (72 h). Cytotoxicity was an unavoidable problem for most cationic polymeric vectors [32] as it was reported that cationic polymers could damage extra cellular matrix proteins and cell membranes [33], leading to impaired cellular uptake of NPs. However, some researchers pointed out that the cellular uptake of cationic polymer NPs was not dependent on their cytotoxicity [34].

### 2.3. Effect of Different DS on Cellular Uptake

Hydrophobized polysaccharide nanoparticles have high stability, the polymer chains form self-aggregates by cross-linking of molecules when the solution concentration is higher than the critical aggregation concentration (CAC) [35]. It should be noted that a concentration of 50 μg/mL is below the lowest CAC of the PEA3 conjugate (0.10 mg/mL), indicating that the PEA conjugates does not form nanoparticles at this concentration. The purpose of the cytotoxicity experiment was to demonstrate the safety of the polymeric material over a wide range of concentrations, while the other three larger concentration gradients focused on the toxic effects of particle size and high concentration on the cells. On the whole, the inhibition of cell growth by high-concentration materials still exists, however, PEA nanoparticles are considered to be non-toxic to cells within a certain range.

Therefore, we selected other three larger concentrations of nanoparticles for further cell uptake experiments. The cellular uptake of FITC-PEA by Caco-2 cells was measured by fluorescence microplate reader. PEA1 was labeled with FITC and the degree of labeling was calculated to be 2.87%. Cellular uptake of PEA-NPs with different DS (PEA1, DS = 3.49; PEA2, DS = 4.18; PEA3, DS = 4.67) at 250, 500, and 1000 μg/mL was evaluated and the results were shown in Figure 3. As can be seen, at the same concentration, cellular uptake efficiency of PEA-NPs increased with increasing DS.

### 2.4. Effect of Different Concentration on Cellular Uptake

The cellular uptake efficiency decreased with increasing concentration of PEA-NPs (Figure 3). This suggested that the uptake process of PEA-NPs was saturable. However, the absolute quantity of PEA-NPs taken into the Caco-2 cells increased with increasing concentration of PEA-NPs as shown in Figure 4. This might be explained that higher concentration of NPs made more PAE carriers engaged in the transportation of the NPs, leading to increased absolute uptake quantity.

### 2.5. Effects of External Environment (Time, Temperature) on Cellular Uptake

To investigate the impacts of incubation time on Caco-2 cellular uptake, PEA3-NPs were incubated with Caco-2 cells at the concentration of 500 μg/mL for 1 h, 2 h, 4 h. Then, the fluorescence intensity was measured and cellular uptake efficiency was calculated (Figure 5A).

To evaluate the effect of incubation temperature on Caco-2 cellular uptake, PEA3-NPs were incubated with Caco-2 cells at 4 °C or 37 °C for 4 h. The cellular uptake efficiency was calculated and shown in Figure 5B. The cellular uptake efficiency at 37 °C was greatly increases by 4.46 folds compared with that at 4 °C.

### 2.6. Cellular Uptake of Alg-Based NPs with Different Hydrophobic Modification

FITC labeled OAE-NPs and PEA3-NPs (500 μg/mL) with the same particle size (200 nm) were incubated with Caco-2 cells for 1 h, 2 h or 4 h before CLSM observation. As shown in Figure 6, the cellular uptake of PEA-NPs exhibited obvious time-dependent manner within 4 h. This result was in accordance with the quantitative data obtained from fluorescence microplate reader as shown in Figure 5A. Similar pattern was observed in OAE-NPs groups. By comparing cellular uptake of PEA-NPs and OAE-NPs, it was shown that the fluorescence intensity in PEA-NPs was higher than that in OAE-NPs at each time point.

### 2.7. Cellular Uptake Profile of PEA-NPs

Nowadays, successful application of NPs in disease diagnosis or treatment requires efficient cellular uptake. It was reported that there were carriers that could recognize and transport phenylalanine on Caco-2 cell membrane [35,36]. The mechanism involved in phenylalanine transport across Caco-2 cell membrane had been well elucidated [37,38]. Cellular uptake of PEA-NPs was an energy-dependent and saturable process with three possible cellular transport routes as depicted in Figure 7.

## 3. Discussion

In this study, cytotoxicity and cellular uptake profile of Alg-based NPs were evaluated in Caco-2 epithelial cells. We found that PEA-NPs did not inhibit proliferation of Caco-2 cells even at the concentration of 1000 μg/mL. Therefore, the effect of the cytotoxicity of the PEA-NPs on Caco-2 cells was disregarded in the subsequent transportation studies [39]. However, we found that higher concentrations of nanoparticles (2000 μg/mL) are toxic to cells over a long period of time (72 h). We speculated that increased concentration of NPs may cause apoptosis in cells, however, the mechanism of nanoparticle toxicity is still unclear. It may be caused by the combination of higher concentrations of nanoparticles and cell membranes for a long time, causing oxidative stress and producing more single-line oxygen or oxidizing active substances, needing to continue to explore and observe in future research [40].

Cellular uptake of PEA-NPs was DS, concentration, time and temperature dependent. Higher cellular uptake efficiency could be achieved using PEA-NPs with higher DS (DS = 4.67) at higher concentration (500 μg/mL) at 37 °C for 4 h. Higher DS of PEA conjugate meant that more PAE were conjugated onto Alg and this might lead to two effects. First, the size of PEA-NPs decreased from 425–227 nm with the higher DS of PAE groups. Smaller size might result in larger number of nanoparticles interacting with cell membrane area, and consequently higher efficiency of cellular uptake [41]. Second, with higher DS of PEA conjugate, there might be more contact with PAE carriers on cell membrane. On the contrary, lower DS of PEA conjugate would have larger particle sizes, which hamper effective binding to receptors due to steric hindrances [42].

Our results showed that cellular uptake efficiency increased with extended incubation time. The cellular uptake efficiency increased sharply when incubation time changed from 1 h to 2 h. However, a milder change was observed when incubation time was prolonged to 4 h. This suggested that when time was extended to as long as 4h, the concentration of PEA-NPs inside the cell got close to the maximum amounts of PEA-NPs that a Caco-2 cell could uptake. In short, the numbers and carrying abilities of PAE carriers on Caco-2 cell membrane was limited. When time increased, the carriers became more close to the saturated state and therefore the cellular uptake efficiency decreased [43]. Therefore, the mechanisms in PEA-NPs uptake, which contain phenylalanine ester in its structure, might also include carrier-mediated transport [44].

The uptake of the PEA-NPs was a process which was dominated by the cells and the process was energy-dependent [45,46]. To be specific, energy could not be supplied for active transportation across the cell membrane at 4 °C, resulting in disturbed cellular uptake compared with that at 37 °C. In contrast, passive transportation is not energy-dependent and consequently irrelevant to temperature. This temperature-dependent uptake manner of PEA-NPs suggested an active transportation process in Caco-2 cells, which was agreed with other cellular uptake results using phenylalanine-based nanoparticles and conjugates [47]. The data about the effects of external environment on the cellular uptake efficiency provided more information on how to achieve the best therapeutic efficacy.

## 4. Materials and Methods

### 4.1. Materials

Phenylalanine ethyl ester (PAE) and oleic acid (OA) was obtained from Solarbio Co. (Beijing, China). The molecular weight of PAE is 229.7 Da, and its purity is more than 99%. The molecular weight of OA is 282.5 Da, and its density is 0.891 g/mL at 25 °C. Alginate (MW = 1.17 × 10^5^ Da, M/G = 1.56) was purchased and measured according to our previous work [47]. Dulbecco’s modified Eagle medium (DMEM) was obtained from HyClone (Logan, UT, USA). Fetal bovine serum (FBS) was obtained from Biochrom AG (Berlin, Germany). Fluorescein isothiocyanate (FITC), 3-(4,5-dimethylthiazol-2-yl)-2,5-diphenyltetrazolium bromide (MTT) and dialysis tube (Mr = 8,000 ~ 14,000 D) were purchased from Sigma Chemical Co. (St. Louis, MO, USA). All other chemicals were of analytical grade and commercially obtained from local chemical suppliers.

### 4.2. Preparation and Characterization of PEA-NPs and OEA-NPs

PEA and OEA were synthesized according to our previous work [48]. To prepare PEA-NPs and OEA-NPs, the PEA and OEA conjugates were dispersed in PBS (15%, *w*/*v*, pH = 7.4) and then sonicated three times with a probe-type sonifier (Sonics Ultrasonic Processor, VC750, Sonics, Hartford, CT, USA) at 90 W for 2 min each [43]. Particle size and size distribution were measured with a Malvern Zetasizer (Malvern Co., Worcestershire, UK) by dynamic light scattering (DLS) [49]. Transmission electron microscope (TEM) (H-600A, Hitachi, Tokyo, Japan) was used to observe the morphology of PEA-NPs.

### 4.3. Cytotoxicity Evaluation

The human colon adenocarcinoma cell line (Caco-2) was purchased from the Center for Excellence in Molecular Cell Science (CAS, Shanghai, China). Cells were cultured in DMEM supplemented with 10% heated-inactivated FBS, 100 U/mL penicillin, 100 μg/mL streptomycin and 2 mmol/L glutamine. Cells were incubated at 37 °C in a humidified atmosphere of 5% CO_2_. Cell viability was determined to assess the in vitro cytotoxicity of PEA-NPs to Caco-2 cells with MTT assay [39]. The Caco-2 cells were seeded onto 96-well plates at a density of 3 × 10^4^ cells/well and cultivated in the incubator to adhere. The culture medium was substituted by medium containing PEA-NPs with different DS at the final concentrations of 50–1000 μg/mL. After incubation for 24 h, MTT solution (0.5 mg/mL) was added to each well and incubated for 4 h at 37 °C. Subsequently, the medium was removed and any formazan crystals formed were solubilized with dimethyl sulfoxide under gentle shaking for 10 min. The absorbance of each well was measured at 490 nm with a Bio-Rad Microplate Reader (Los Angeles, CA, USA). Each concentration of the samples had six replicates. Every test includes a blank containing culture medium only. The viability was presented as the percent of sample well to the control well.

### 4.4. Labeling of NPs with Fluorescein Isothiocyanate (FITC)

The fluorescent labeling of PEA to study cell uptake profile of PEA-NPs was adapted from previously reported methods with minor modifications [45]. Briefly, 500 μL of FITC-methanol solution (1 mg/mL) was mixed with 10 mL PEA conjugates dissolved in PBS (15%, *w*/*v*). After reaction in the dark with shaking at room temperature for 3 h, the solution was precipitated with excess 95% ethanol overnight. After centrifugation (6000 g, 40 min), the precipitant was collected and re-dissolved in methanol (1 mg/mL). The labeling efficiency of FITC was determined. Fluorescence intensity of the FITC labeled-PEA (FITC–PEA) was measured on a fluorescence spectrophotometer (RF-5301PC, Shimadzu, Kyoto, Japan). The weight of FITC in the FITC–PEA could be calculated from the FITC standard curve which was depicted in advance. The degree of labeling (percentage) was calculated as the percent weight of FITC to the weight of FITC–PEA using the following equation:(1)$$\mathrm{Degree}\;\mathrm{of}\;\mathrm{labeling}\;(\%)\;=\;\frac{\mathrm{FITC}\;\left(\mathrm{mg}\right)}{\mathrm{FITC}-\mathrm{PA}\;\left(\mathrm{mg}\right)}\;\;\times \;100\%,\;$$

### 4.5. Cellular Uptake of FITC-PEA Nanoparticles

Cellular uptake of FITC-PEA NPs was visualized by Confocal Laser Scanning Microscopy (CLSM) and quantified by fluorescence intensity given by a fluorescence microplate reader (FLx800B, Bio-Tek, Burlington, VT, USA). Caco-2 cells were seeded into 96-well plates (8 × 10^3^ cells/well) for cellular uptake efficiency evaluation and cells were also plated into 6-well plates (1 × 10^5^ cells/mL) for CLSM observation. After 24 h of incubation, FITC-PEA NPs were added into the cells in DMEM without FBS at predestined temperature (4 °C and 37 °C) for predetermined time (1, 2, and 4 h). After washing twice with PBS and fixing with 4% paraformaldehyde solution for 10 min at 4 °C, the cells were visualized with CLSM (Leica TCS SP5, Solms, Germany). For excitation of FITC fluorescence, an argon laser with an excitation wavelength of 488 nm was used. And the emission wavelength was 512 nm. The fluorescence intensity of FITC was quantified with fluorescence microplate reader (FLx800B, Bio-Tek) after cytolysis with 0.5% Triton X-100 for 15 min. Cellular uptake efficiency was calculated using the following equations, where *Is* represents the fluorescence intensity of sample groups after co-incubation, *In* represents the fluorescence intensity of negative control group (cells incubated without any FITC-PEA particles), and *Ip* represents the fluorescence intensity of sample groups before co-incubation:(2)$$\mathrm{Cellular}\;\mathrm{uptake}\;\mathrm{efficiency}\;=\;\frac{I_{s}-I_{n}}{I_{p}-I_{n}\;}\;\times \;100\%,$$

### 4.6. Statistical Analysis

All data were expressed as mean ±SD. Data were analyzed statistically by the SPSS 11.5 software package (IBM, Armonk, NY, USA). * *P* < 0.05 was regarded as significant level and ** *P* < 0.01 was considered as very significant level.

## 5. Conclusion

In conclusion, our study found that PEA-NPs were bio-safe and that had many exciting properties, such as stability in different solutions and good degradability after uptake by cells. PAE modified alginate nanoparticles had stronger cell permeability than OA modified alginate nanoparticles. Higher cell permeability suggested that PEA-NPs might be absorbed more easily by the small intestinal cells. It showed the strong permeability of PEA-NPs across human intestinal barrier and provided foundation for application of PEA-NPs as oral drug or nutrient delivery system. Cellular uptake of PEA-NPs was an energy-dependent and saturable process with three possible cellular transport routes, however, the detailed uptake mechanisms and pathways still need more in-depth research [50]. In the future, clathrin, caveolae and macropinocytosis inhibitors could be used in further studies to elucidate the specific transport mechanism of PEA-NPs [51,52], and intestinal version model could be used to further study the cellular uptake and transportation behavior to provide more information for the application of PEA-NPs as nutrient or drug delivery system [28].

## Figures and Tables

**Figure 1 molecules-24-00555-f001:**
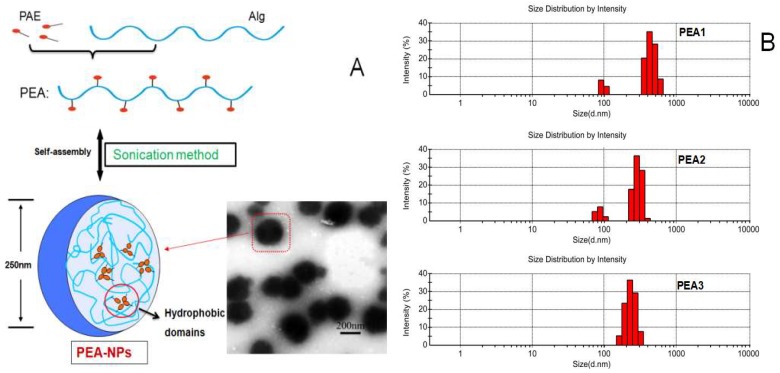
(**A**): Transmission electron microscopy (TEM) image of PEA-NPs based on Alg; (**B**): Size distribution measured by DLS.

**Figure 2 molecules-24-00555-f002:**
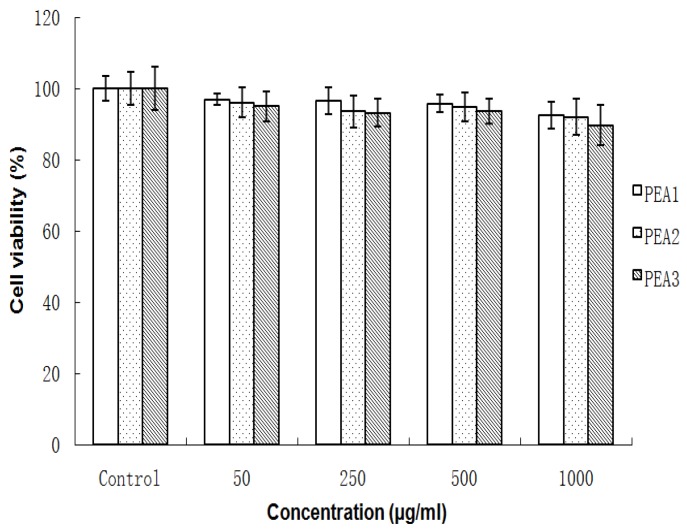
Viability of Caco-2 cells treated with the PEA1, PEA2, PEA3 nanoparticles at various concentrations for 48 h. Data are expressed as mean values (with standard deviation) of six experiments.

**Figure 3 molecules-24-00555-f003:**
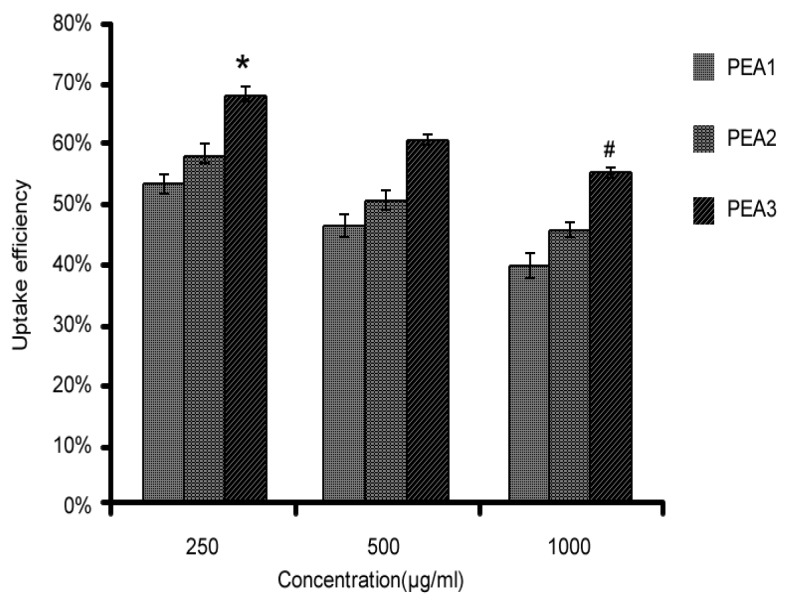
Cellular uptake of different PEA-NPs by Caco-2 cells at various concentrations for 4 h. Note: * represents significance among various DS groups at the same concentration, and # represents significance among various concentration at the same DS.

**Figure 4 molecules-24-00555-f004:**
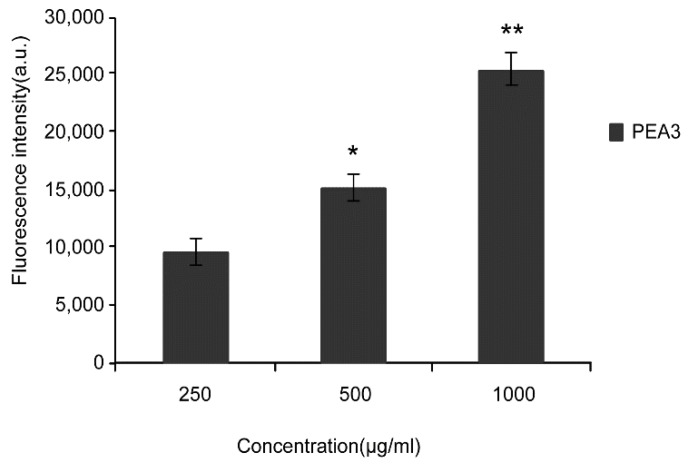
Fluorescence intensity of PEA3-NPs by Caco-2 cells at various concentrations for 4 h. Note: * *P* < 0.05, ** *P* < 0.01 significant difference compared with control groups (250 μg/mL).

**Figure 5 molecules-24-00555-f005:**
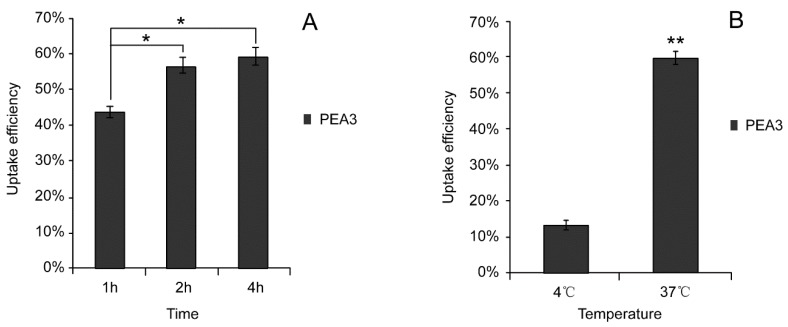
Cellar uptake efficiency of PEA3 nanoparticles by Caco-2 cells at (**A**) various times and (**B**) various temperatures for 4 h. Note: * *P* < 0.05, ** *P* < 0.01 significant difference compared with control groups (1 h, 4 °C).

**Figure 6 molecules-24-00555-f006:**
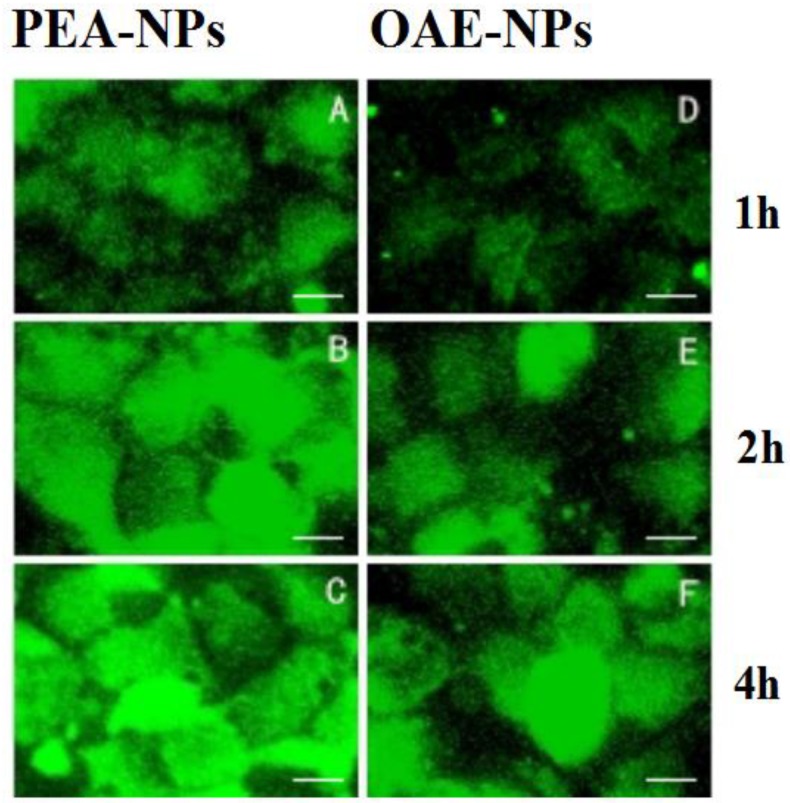
CLSM images of Caco-2 cells treated with FITC labeled PEA-NPs or OAE-NPs: 1 h (**A**,**D**), 2 h (**B**,**E**) and 4 h (**C**,**F**); scale bar represents 25 μm.

**Figure 7 molecules-24-00555-f007:**
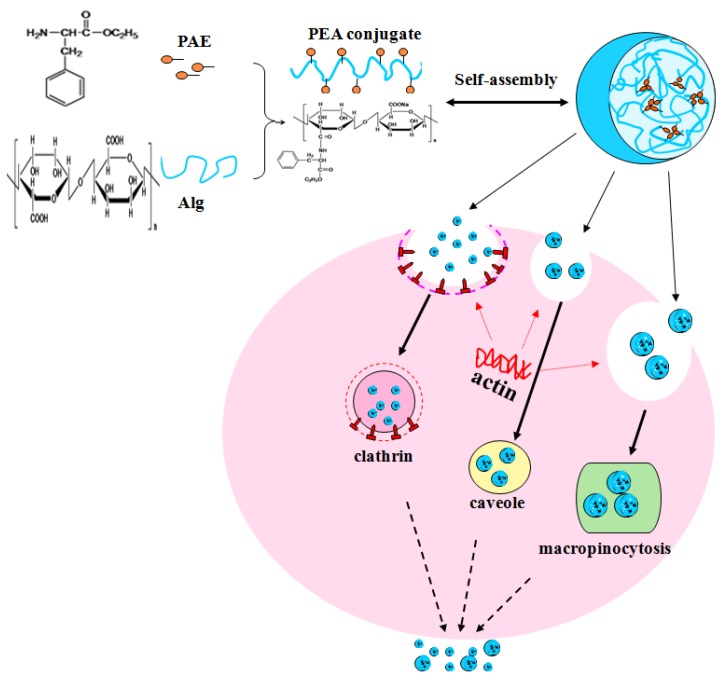
Schematic diagram of self-assembly of PEA-NPs in distilled water and three energy-dependent transport manners in Caco-2 epithelial cells.

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
