# Peer review of "Biocompatibility Profile and In Vitro Cellular Uptake of Self-assembled Alginate Nanoparticles"

_molecules, 2019, doi:10.3390/molecules24030555_

Round 1
Reviewer 1 Report
In this manuscript, the authors prepared the phenylalanine ethyl ester (PAE) modified alginate (Alg) nanoparticles and evaluated the biocompatibility of the nanoparticles. Overall, this work fits the scope of the journal and the results are clear. However, several issues and details need to be addressed.
1. To enhance significance and novelty of this work, I recommend the authors change a little bit organization in the introduction part by introducing the nanoparticle applications in different fields, such as cancer therapy.
2. The molecular weight of PAE should be measured and provided.
3. As an important property, the stability of the nanoparticles should be evaluated. I recommend the authors evaluate the nanoparticle stability in both PBS and cell culture medium and the nanoparticle size should be measured at different time points.
4. The authors should investigate the subcellular localization of nanoparticles by labeling the endosomes/lysosomes, nuclei and nanoparticles with the fluorescent dye.
Author Response
Response to Reviewer 1 Comments
Point 1: To enhance significance and novelty of this work, I recommend the authors change a little bit organization in the introduction part by introducing the nanoparticle applications in different fields, such as cancer therapy.
1. Response 1: We are very grateful to the reviewer for the content of introducing the nanoparticle applications in different fields which was neglected in this manuscript. We added the following examples in the introduction:Wang have investigated OAlg-based nanoparticles in the self-assembled films not only result in porous nanostructures similar to natural ECM, but also preserve the activity and realize the sustained release of Bone morphogenetic protein-2(Wang Z , Dong L , Han L , et al. Self-assembled Biodegradable Nanoparticles and Polysaccharides as Biomimetic ECM Nanostructures for the Synergistic effect of RGD and BMP-2 on Bone Formation[J]. Scientific Reports, 2016, 6:25090). Kolya have prepared gold nanoparticles as antibacterial agents by using graft copolymer based on sodium alginate, dimethylacrylamide (DMA) and acrylic acid (AA)(Kolya H , Pal S , Pandey A , et al. Preparation of gold nanoparticles by a novel biodegradable graft copolymer sodium alginate-g-poly (N,N-dimethylacrylamide-co-acrylic acid) with anti micro bacterial application[J]. European Polymer Journal, 2015, 66:139-148). Yu have prepared the photosensitizer- encapsulated amphiphilic sodium alginate derivative nanoparticles to enhance phototoxicity in the photodynamic therapy of pancreatic cancer(Yu Z , Li H , Zhang L M , et al. Enhancement of phototoxicity against human pancreatic cancer cells with photosensitizer-encapsulated amphiphilic sodium alginate derivative nanoparticles[J]. International Journal of Pharmaceutics, 2014, 473(1-2):501-509). Song fabricated the magnetic alginate/CHI layer-by-layer nanoparticles (MACPs) to deliver curcumin for treating with breast cancer cells(Song W , Su X , Gregory D , et al. Magnetic Alginate/Chitosan Nanoparticles for Targeted Delivery of Curcumin into Human Breast Cancer Cells[J]. Nanomaterials, 2018, 8(11)).
Point 2: The molecular weight of PAE should be measured and provided.
Response 2: We are very sorry that we forgot to provide basic information about the materials. The molecular weight of PAE is 229.7 Da,and its purity is more than 99%. The molecular weight of OA is 282.5 Da, and its density is 0.891g/mL at 25°C. Sodium alginate (MW≈1.17×105 Da;viscosity≈225 cp,2% (25°C) (lit.);M/G≈1.56 (measured according to our previous work).
Point 3: As an important property, the stability of the nanoparticles should be evaluated. I recommend the authors evaluate the nanoparticle stability in both PBS and cell culture medium and the nanoparticle size should be measured at different time points.
Response 3: We agree with the comment and understand that the stability of the nanoparticles is an important property. In our previous study, we successfully prepared nanoparticles based on alginic acid and uesd OAE-NPs as Carriers for Sustained Release of Vitamin D3.The effect of different time and solutions on stability of OAE Nanoparticles is detected (Li Q , Liu C G , Huang Z H , et al. Preparation and Characterization of Nanoparticles Based on Hydrophobic Alginate Derivative as Carriers for Sustained Release of Vitamin D3[J]. Journal of Agricultural & Food Chemistry, 2011, 59(5):1962). In our previous article, PEA-3 nanoparticles (PEA3-NPs) among these samples, which had the highest degree of PAE substitution (DS=4.67%), possessed the smallest particle size (226.7 ± 2.79 nm) and the maximum VB2 loading capacity (LC, 3.53 ± 0.03%) in PBS solution.These results strongly support the conclusion that nanoparticles based on PEA conjugates could be potentially used as a nanosized NDS with good stability(Zhang P , Zhao S R , Li J X , et al. Nanoparticles based on phenylalanine ethyl ester-alginate conjugate as vitamin B2 delivery system[J]. Journal of Biomaterials Applications, 2016:0885328216630497).We are very sorry that we have neglected to explain this problem and have marked it in the original revised manuscript.
Point 4: The authors should investigate the subcellular localization of nanoparticles by labeling the endosomes/lysosomes, nuclei and nanoparticles with the fluorescent dye.
Response 4: Thank you for these important suggestions.We agree with the comment and understand that the subcellular localization of nanoparticles by labeling the endosomes/lysosomes, nuclei and nanoparticles with the fluorescent dye may better reveal the transport and localization of nanoparticles at the subcellular level. However, in the present study, we mainly focused on the biocompatibility and cellular uptake profiles of this nanocarrier, and we think that cytocompatibility of PEA-NPs and effect of different DS or external environment on cellular uptake may not be optimal, but should be sufficient to draw a conclusion that PEA-NPs were nontoxic and cellular uptake of PEA-NPs was DS, concentration, time and temperature dependent. The subcellular localization of nanoparticles is under investigation in our laboratory. Unfortunately, results are unavailable at this point.
The authors are grateful to the referee for pointing out their error.
Special thanks to you for your good comments!

Reviewer 2 Report
Zhang et al. investigate that PEA-NPs has not induced any effects on the proliferation of the human intestinal epithelial Caco-2 cells at concentrations up to 1000 μg/ml. Moreover, the in vitro cellular uptake profile of PEA-NPs reveal that these NPs could potentially be used for the disease diagnosis and treatment. This is an interesting study. However, there is a lot of unclear mechanism explanation in the strategy. For example, in this manuscript, there is no content related to the mechanism of cell death. Secondly, there is no comparison with conventionally used polymeric-NPs at different molecular levels. Authors may provide justification why they chose these NPs. Based on these concerns, I recommend this manuscript be accepted with subject to major revisions.
Following are the specific changes I can suggest;
- Page 1 line 31: what are ‘and other biomedical applications’?
- Page 2, line 46, ‘therapeutics’? The sentence does not make sense. Please revise this sentence.
- Page 2, line 49, the word ‘proper’ can be replaced by ‘standard’
- Page 2, line 50, ‘We previous reported’ is incorrect.
- Page 2, line 56, ‘We wish that’ is not a scientific language.
- Figure 1 b is not readable. The quality of the image can be improved.
- What is effect of surface charge on toxicity in a pH dependent manner? What is the pH of cells when used for toxicity screening?
- What is cell death mechanism? It is better to discuss around oxidative stress induced by these NPs? Discuss around the paper ‘Tabish, T. A., Zhang, S., & Winyard, P. G. (2018). Developing the next generation of graphene-based platforms for cancer therapeutics: the potential role of reactive oxygen species. Redox biology, 15, 34-40’.
- Reactive oxygen species production genuinely contribute to DNA cleavage. It is better to discuss about the toxic stress that generates oxidative DNA damage.
- Can these nanostructure cross blood brain barrier which is very important for diagnosis and treatment of medical conditions?
- How about the uptake and fluorescence intensity of 50 μg/ml.
- The quality of figure 6 is very poor. The scale bar must be added within the images.
- How about the biodegradability of these NPs?
- There are several typo mistakes and inconsistent verb tense.
Following reference can be added;
Song, W., Su, X., Gregory, D., Li, W., Cai, Z., & Zhao, X. (2018). Magnetic Alginate/Chitosan Nanoparticles for Targeted Delivery of Curcumin into Human Breast Cancer Cells. Nanomaterials, 8(11), 907.
Brezaniova, I., Trousil, J., Cernochova, Z., Kral, V., Hruby, M., Stepanek, P., & Slouf, M. (2017). Self-assembled chitosan-alginate polyplex nanoparticles containing temoporfin. Colloid and Polymer Science, 295(8), 1259-1270.
Quiñones, J. P., Peniche, H., & Peniche, C. (2018). Chitosan based self-assembled nanoparticles in drug delivery. Polymers, 10(3), 235.
Tabish TA, Scotton C, Ferguson D, Liangxu L, Veen A, Lowry S, Ali M, Jabeen F, Ali M, Winyard P. (2018) Biocompatibility and toxicity of graphene quantum dots for potential application in photodynamic therapy, Nanomedicine, pages 1-16, DOI:10.2217/nnm-2018-0018
Liu, Y., Kong, M., Feng, C., Yang, K. K., Li, Y., Su, J., & Chen, X. G. (2013). Biocompatibility, cellular uptake and biodistribution of the polymeric amphiphilic nanoparticles as oral drug carriers. Colloids and Surfaces B: Biointerfaces, 103, 345-353.
Wang, Z., Dong, L., Han, L., Wang, K., Lu, X., Fang, L., & Chan, C. W. (2016). Self-assembled Biodegradable Nanoparticles and Polysaccharides as Biomimetic ECM Nanostructures for the Synergistic effect of RGD and BMP-2 on Bone Formation. Scientific reports, 6, 25090.
Author Response
Response to Reviewer 2 Comments
Point 1: In this manuscript, there is no content related to the mechanism of cell death.
Response 1: We are very grateful to the reviewer for the content of cell death which was neglected in this manuscript. The general definition of Cell death is the irreversible stopping of cellular life activities. Apoptosis and necrosis are two distinct types of cell death, characterized by differences in cellular morphology and executive molecules(Degterev A , Huang Z , Boyce M , et al. Chemical inhibitor of nonapoptotic cell death with therapeutic potential for ischemic brain injury[J]. Nature Chemical Biology, 2005, 1(2):112-119).
Necrosis is thought to be a passive cell death. Apoptosis, in contrast, is a controlled biological process to remove unwanted cells and, thus, is regulated in a sophisticated manner with antiapoptotic molecules, such as Bcl-2 and Bcl-XL , or proapoptotic ones, such as Bax and Bak. During the processes of apoptosis, cells undergo distinct biochemical and morphological changes, which include membrane blebs and shrinking, nuclear condensation, DNA fragmentation, and disintegration of the dying cell into apoptotic bodies(Huang, Jiang G R , Wu S , et al. Induction of cell death of gastric cancer cells by a modified compound of the annonaceous acetogenin family[J]. Chembiochem, 2010, 4(11):1216-1221).The mechanism of nanoparticle toxicity is still unclear. Apoptosis or autolysis of cells may be caused by different surface modifications, different nanometer particle sizes, and different modes of action.
Point 2: There is no comparison with conventionally used polymeric-NPs at different molecular levels. Authors may provide justification why they chose these NPs.
Response 2: The reasons for choosing PEA-NPs as the sample tested are as following:
(1) In our previous study, We successfully prepared nanoparticles based on carboxymethyl chitosan(MW≈1.66×104 Da)(Tan Y L , Liu C G . Preparation and characterization of self-assemblied nanoparticles based on folic acid modified carboxymethyl chitosan[J]. Journal of Materials Science: Materials in Medicine, 2011, 22(5):1213-1220), hyaluronic acid(MW≈1.66×104 Da)(Dong X, Liu C. Preparation and characterization of self-assembled nanoparticles of hyaluronic acid-deoxycholic acid conjugates[J]. Journal of Nanomaterials. 2010.) and alginic acid(MW≈1.17×105 Da)(Li Q , Liu C G , Huang Z H , et al. Preparation and Characterization of Nanoparticles Based on Hydrophobic Alginate Derivative as Carriers for Sustained Release of Vitamin D3[J]. Journal of Agricultural & Food Chemistry, 2011, 59(5):1962). Alginate modified with oleic acid enhanced the permeability of an insoluble vitamin through intestine cell membrane (Li Q, Liu CG and Yu Y. Separation of monodisperse alginate nanoparticles and effect of particle size on transport of vitamin E. Carbohydr Polym. 2015; 124: 274–9). So the proposal of the research was that the PEA conjugates could be self-assembled into nanoparticles and used as a carrier for drugs.
(2) Among these samples in our previous article, PEA-3 nanoparticles (PEA3-NPs), which had the highest degree of PAE substitution (DS=4.67%), possessed the smallest particle size (226.7 ± 2.79 nm) and the maximum VB2 loading capacity (LC, 3.53 ± 0.03%) and loading efficiency (LE, 91.48 ± 0.80%).These results strongly support the conclusion that nanoparticles based on PEA conjugates could be potentially used as a nanosized NDS(Zhang P , Zhao S R , Li J X , et al. Nanoparticles based on phenylalanine ethyl ester-alginate conjugate as vitamin B2 delivery system[J]. Journal of Biomaterials Applications, 2016:0885328216630497).
Point 3: Page 1 line 31: what are ‘and other biomedical applications’?
Response 3: According to the comments, “and other biomedical applications” has been corrected:“and other biomedical applications,such as suitable properties for potential use PDT4换2017, affecting immune responses5,6,delivery of proteomics or genomics used in ligand-targeted therapeutics7,delivery of anticancer and antifungal drugs approved for clinical use8” .
Point 4: Page 2, line 46, ‘therapeutics’? The sentence does not make sense. Please revise this sentence.
Response 4: According to the comments, the word “therapeutics” has been replaced:"effectiveness".
Point 5: Page 2, line 49, the word ‘proper’ can be replaced by ‘standard’
Response 5: According to the comments, the word “proper” has been replaced:" standard".
Point 6: Page 2, line 50, ‘We previous reported’ is incorrect.
Response 6: According to the comments, “We previous reported” has been corrected:" In a previous study, we reported". We are very sorry for our incorrect writing.
Point 7: Page 2, line 56, ‘We wish that’ is not a scientific language.
Response 7: According to the comments, “We wish that” has been corrected:" It suggest that".
Point 8: Figure 1 b is not readable. The quality of the image can be improved.
Response 8: According to the comments,we have improved the quality of Figure 1B.
Point 9: What is effect of surface charge on toxicity in a pH dependent manner? What is the pH of cells when used for toxicity screening?
Response 9: We are very sorry that we forgot to provide the pH of cells when used for toxicity screening. The pH is 7.4 for the reason that the pH of Dulbecco’s modified Eagle medium (DMEM) which is used to incubate Cells is 7.4. To prepare PEA-NPs and OEA-NPs, the PEA and OEA conjugates were dispersed in PBS (15%, W/V,pH=7.4) and then sonicated three times with a probe-type sonifier. Thus, the pH of NPs is same as the DMEM’s .
The mechanism of nanoparticle toxicity is still unclear. Apoptosis or autolysis of cells may be caused by different surface modifications, different nanometer particle sizes, and different modes of action. Changes in surface charge may increase the toxicity of nanoparticles and even cause apoptosis in cells. (Huang, Jiang G R , Wu S , et al. Induction of cell death of gastric cancer cells by a modified compound of the annonaceous acetogenin family[J]. Chembiochem, 2010, 4(11):1216-1221).
Point 10: What is cell death mechanism? It is better to discuss around oxidative stress induced by these NPs? Discuss around the paper ‘Tabish, T. A., Zhang, S., & Winyard, P. G. (2018). Developing the next generation of graphene-based platforms for cancer therapeutics: the potential role of reactive oxygen species. Redox biology, 15, 34-40’.
Response 10: According to the comments, we have added the cell death mechanism to the discussion of revised manuscript, as shown in point 1. “Apoptosis is a controlled biological process to remove unwanted cells and, thus, is regulated in a sophisticated manner with antiapoptotic molecules, such as Bcl-2 and Bcl-XL , or proapoptotic ones, such as Bax and Bak. During the processes of apoptosis, cells undergo distinct biochemical and morphological changes, which include membrane blebs and shrinking, and DNA fragmentation.”We agree with the comment and understand that the inherent toxicity of NPs, which is a pivotal consideration for future therapeutic application. In this study, We found that PEA-NPs did not inhibit proliferation of Caco-2 cells at the concentration of 1000μg/mL. However, high concentrations of nanoparticles are toxic to cells over a long period of time(72 h). We speculated that increased concentration of NPs may cause apoptosis in cells,however, the mechanism of nanoparticle toxicity is still unclear. Apoptosis or autolysis of cells may be caused by the combination of higher concentrations of nanoparticles and cell membranes for a long time, causing oxidative stress and producing more single-line oxygen or oxidizing active substances, needing to continue to explore and observe in future research.
Point 11: Reactive oxygen species production genuinely contribute to DNA cleavage. It is better to discuss about the toxic stress that generates oxidative DNA damage.
Response 11: Thank you for this important suggestion.We agree with the comment and understand that the toxic stress that generates oxidative DNA damage may better reveal the cell death mechanism. During the processes of apoptosis, cells undergo distinct biochemical and morphological changes, which include membrane blebs and shrinking, nuclear condensation and DNA fragmentation. However, in the present study, we mainly focused on the biocompatibility and cellular uptake profiles of this nanocarrier, and we think that cytocompatibility of PEA-NPs and effect of different DS or external environment on cellular uptake may not be optimal, but should be sufficient to draw a conclusion that PEA-NPs were nontoxic and cellular uptake of PEA-NPs was DS, concentration, time and temperature dependent. The DNA damage generated by toxic stress is under investigation in our laboratory. Unfortunately, results are unavailable at this point.”
Point 12: Can these nanostructure cross blood brain barrier which is very important for diagnosis and treatment of medical conditions?
Response 12: The blood-brain barrier can protect the brain, but it also prevents approximately 98% of small molecule drugs and almost all macromolecular compounds from entering the brain's blood circulation. Nowadays,the study of drug-loaded nanoparticles across the blood-brain barrier has received more and more attention for the reason that nanoparticles prepared by some special materials can effectively increase the amount of drugs passing through the blood-brain barrier.
The possible mechanism of the nanoparticles penetrating the BBB system is as follows: 1.Adsorption of nanoparticles on the capillary wall of the brain makes the drug more easily enter the brain through the capillary endothelium; 2.Interactions open the tight junctions between nanoparticles and BBB capillary endothelial cells; 3. Capillary endothelial cells can swallow the nanoparticles by phagocytosis, allowing the drug to release and spread into the brain.
However, the particle size of the PEA nanoparticles prepared in this paper is between 220 and 430 nm, which cannot directly penetrate the blood-brain barrier system into the brain, but this is one of the main directions for our next experiments.
Point 13: How about the uptake and fluorescence intensity of 50 μg/ml.
Response 13: Hydrophobized polysaccharide nanoparticles have high stability, the polymer chains form self-aggregates by cross-linking of molecules when the solution concentration is higher than the critical aggregation concentration(CAC). It should be noted that a concentration of 50 μg/ml is below the lowest CAC of the PEA3 conjugate(0.10mg/ml), indicating that the PEA conjugates does not form nanoparticles at this concentration. The purpose of the cytotoxicity experiment was to demonstrate the safety of the polymeric material over a wide range of concentrations, while the other three larger concentration gradients focused on the toxic effects of particle size and high concentration on the cells. On the whole, the inhibition of cell growth by high-concentration materials still exists, however, PEA nanoparticles are considered to be non-toxic to cells within a certain range. Therefore, we selected other three larger concentrations of nanoparticles for further cell uptake experiments.
Point 14: The quality of figure 6 is very poor. The scale bar must be added within the images.
Response 14: According to the comments,we have improved the quality of Figure 6, however, the change in the clarity of the image is not very large for the reason of the original photo was not found. At the same time, scale bar have been added within all images which represents 25 μm.
Point 15: How about the biodegradability of these NPs?
Response 15: PEA-NPs have good biodegradability. Firstly, sodium alginate is a biocompatible polysaccharide which is non-toxic to humans, phenylalanine ethyl ester is also a kind of amino acid benzene which constitutes human protein. PEA nanoparticle is easy to degrade because of two natural components. Secondly, the diffusion and biodegradation of drug-loaded nanoparticles are the main factors controlling the drug release process. The sustained release of vitamins from PEA nanoparticles in the early stage proved that PEA drug-loaded nanoparticles have good drug release effect within 2 days. In addition, SP nanoparticles are more effective in degradation and release under neutral or alkaline conditions.
Point 16: There are several typo mistakes and inconsistent verb tense.
Response 16: According to the comments, the word “coincubation” has been replaced:" co-incubation" in Page 8, line 235,
“Clathrin” has been replaced:" clathrin" in Page 8, line 251,
“everted intestinal ring model” has been replaced:" intestinal version model " in Page 8, line 253.
Point 17: Following reference can be added:
Song, W., Su, X., Gregory, D., Li, W., Cai, Z., & Zhao, X. (2018). Magnetic Alginate/Chitosan Nanoparticles for Targeted Delivery of Curcumin into Human Breast Cancer Cells. Nanomaterials,8(11), 907.
Brezaniova, I., Trousil, J., Cernochova, Z., Kral, V., Hruby, M., Stepanek, P., & Slouf, M. (2017). Self-assembled chitosan-alginate polyplex nanoparticles containing temoporfin. Colloid and Polymer Science, 295(8), 1259-1270.
Quiñones, J. P., Peniche, H., & Peniche, C. (2018). Chitosan based self-assembled nanoparticles in drug delivery. Polymers, 10(3), 235.
Tabish TA, Scotton C, Ferguson D, Liangxu L, Veen A, Lowry S, Ali M, Jabeen F, Ali M, Winyard P. (2018) Biocompatibility and toxicity of graphene quantum dots for potential application in photodynamic therapy, Nanomedicine, pages 1-16, DOI:10.2217/nnm-2018-0018
Liu, Y., Kong, M., Feng, C., Yang, K. K., Li, Y., Su, J., & Chen, X. G. (2013). Biocompatibility, cellular uptake and biodistribution of the polymeric amphiphilic nanoparticles as oral drug carriers.Colloids and Surfaces B: Biointerfaces, 103, 345-353.
Wang, Z., Dong, L., Han, L., Wang, K., Lu, X., Fang, L., & Chan, C. W. (2016). Self-assembled Biodegradable Nanoparticles and Polysaccharides as Biomimetic ECM Nanostructures for the Synergistic effect of RGD and BMP-2 on Bone Formation. Scientific reports, 6, 25090.
Response 17: According to the comments,we have added all of above references in this manuscript (in red).
Once again, thank you very much for your comments and suggestions.

Round 2
Reviewer 1 Report
No further comments.
Reviewer 2 Report
The authors have successfully addressed all remaining points. The presentation now provides sufficient clarity, and I have confidence that the data analysis is valid.
I therefore have no further objections against publication of the manuscript in Molecules.